# Orientation Dependence of Cathodoluminescence and Photoluminescence Spectroscopy of Defects in Chemical-Vapor-Deposited Diamond Microcrystal

**DOI:** 10.3390/ma13235446

**Published:** 2020-11-29

**Authors:** Kazimierz Fabisiak, Szymon Łoś, Kazimierz Paprocki, Mirosław Szybowicz, Janusz Winiecki, Anna Dychalska

**Affiliations:** 1Institute of Physics, Kazimierz Wielki University, Powstańców Wielkopolskich 2, 85090 Bydgoszcz, Poland; kfab@ukw.edu.pl (K.F.); paprocki@ukw.edu.pl (K.P.); 2Faculty of Materials Engineering and Technical Physics, Poznań University of Technology, Piotrowo 3, 60965 Poznań, Poland; anna.dychalska@put.poznan.pl; 3Oncology Center, Medical Physics Department, ul. Romanowskiej 2, 85796 Bydgoszcz, Poland; j.winiecki@cm.umk.pl; 4Chair and Clinic Oncology and Brachytherapy, Nicolaus Copernicus University, 87100 Toruń, Poland

**Keywords:** diamond point defects, cathodoluminescence, Raman spectroscopy, photoluminescence, defects concentration

## Abstract

Point defects, impurities, and defect–impurity complexes in diamond microcrystals were studied with the cathodoluminescence (CL) spectroscopy in the scanning electron microscope, photoluminescence (PL), and Raman spectroscopy (RS). Such defects can influence the directions that microcrystals are grown. Micro-diamonds were obtained by a hot-filament chemical vapor deposition (HF CVD) technique from the methane–hydrogen gas mixture. The CL spectra of diamond microcrystals taken from (100) and (111) crystallographic planes were compared to the CL spectrum of a (100) oriented Element Six diamond monocrystal. The following color centers were identified: 2.52, 2.156, 2.055 eV attributed to a nitrogen–vacancy complex and a violet-emitting center (A-band) observed at 2.82 eV associated with dislocation line defects, whose atomic structure is still under discussion. The Raman studies showed that the planes (111) are more defective in comparison to (100) planes. What is reflected in the CL spectra as (111) shows a strong band in the UV region (2.815 eV) which is not observed in the case of the (100) plane.

## 1. Introduction

Diamond is characterized by a unique combination of physical properties, making it a promising material with possible versatile applications. It is widely known for its hardness. Industrial diamond has been successfully applied to drilling and cutting tools all over the world [1,2]. Diamond is one of the best electrical insulators while being an excellent heat conductor, five times better than that of copper. When doped, for example with boron, it can be considered for use in semiconductor devices [3]. However, an artificially grown diamond can also be used in particle detection, similar to widely used silicon or germanium. Due to its expected radiation hardness, diamond is a candidate for future high energy experiments [4]. However, there still exist some limitations on the performance of diamond electronic device applications. They are because the chemical vapor deposited diamond layers (CVD) contain a high concentration of different defects types. The source of these defects may be the lattice irregularities, vacancies, stacking faults, intentional doping, or interstitial impurities created during the diamond growth process. Optical and optoelectronic applications of CVD-grown diamonds are expected to take advantage of this material’s desirable intrinsic properties [5,6]. The optical properties and electrical transport properties of a wide-bandgap semiconductor, such as the diamond, are susceptible to states within the forbidden gap produced by lattice defects or impurity atoms. For many years, scientists have been looking for ideal candidates to use as information carrier qubits for quantum computation and communications. Nitrogen vacancy centers (NV) in the diamond are among the most interesting objects for producing quantum elements that fulfill the well-established DiVincenzo’s criteria for quantum information technology at room temperature [7]. Especially electron spins localized at NV centers, at atomic scales they can be manipulated at room temperature by applying a magnetic field, electric field, microwave radiation, or light, or a combination. We believe that single crystallites and polycrystalline diamond layers with a (100) preferential orientation can be used for the above-mentioned applications. This is confirmed by the results published in our previous work [8]. Testing of these defects is necessary and can be done using several types of measurements.

The most effective method to characterize the structural quality of both polycrystalline diamond layers and diamond monocrystals is the commonly used Raman spectroscopy [9]. However, this method does not provide information on the nature of these defects. Moreover, structural defects and dopants introduce disorder in the crystal lattice, and associated energy states inside the diamond energy bandgap can act as color centers responsible for luminescence in visible spectral region [8]. Thus, complementary methods for testing various defect types are photoluminescence (PL) and cathodoluminescence (CL) spectroscopy. In general, for almost all diamond types, the broad “band A” at 440 nm in the cathodoluminescence spectra is observed. In the case of natural diamonds, this band is assigned to donor–acceptor recombination [10]. However, in the undoped CVD diamond, it seems to be related to structural defects like dislocations [11]. Therefore, the use of the CL, PL, and RS combination can be an effective method for determining the quality of diamond crystallinity and the type of defects within the crystal lattice.

In the era of miniaturization, we believe that small-size diamond microcrystallites can be used in optoelectronic devices, which is one of the motivations for undertaking of this type of research.

In this work, we present the investigation of single diamond microcrystal properties grown by hot filament chemical vapor deposition method (HF CVD). It is carried out utilizing combined techniques, i.e., RS, CL, and scanning electron microscopy (SEM). In particular, we will try to find a correlation between the A-band emission with the full width at half maximum of the Raman line (FWHM) to support the presence of dislocations in the crystal lattice.

## 2. Materials and Methods

We chose diamond microcrystallites of different orientations for our research in order to study the differences in the degree of their defectiveness [8] and possible suitability for sensory applications [12]. The properties of diamond microcrystallites obtained by the HF CVD method were compared with those of a single diamond crystal purchased from Element Six.

Deposited single microcrystal diamonds on the (100) oriented silicon substrate were obtained by the HF CVD technique. The apparatus’s reaction chamber consisted of stainless steel with an internal diameter of 50 cm and was cooled by water. A tungsten filament with 2 mm distance from the substrate, heated up to 2100 °C, was used for thermal activation of the working gas mixture of methane and hydrogen (CH_4_/H_2_ = 1 vol.%). The parameters of the growth process were as follows: the total pressure in the reaction chamber—p = 80 mbar, the substrate temperature—1000 K, and the working gas flow rate—100 sccm. The 99.99 purity methane and hydrogen gases were supplied by The Linde Gaz Group company.

Before starting the diamond microcrystal growth process, the Si substrate was washed in acetone and then ethanol in an ultrasonic bath. To grow separate crystals, it was not mechanically polished to avoid creating surface defects, preventing the growth of a continuous polycrystalline layer.

The diamond microcrystal morphology was studied using SEM, Jeol JSM-6300 operating at a voltage of 20 kV. SEM images were also used to examine grain sizes. The CL emission spectra were registered at room temperature using a grating spectrometer equipped with a charge-coupled device camera. The CL signal from SEM to the monochromator was sent by optical fiber. It was recorded by the grating StellarNet’s SILVVER–Nova (StellarNet, Inc. Tampa, FL, USA) high performance fiber optic spectrometer covering the 190–1100 nm wavelength range equipped with a TE cooled 2048 pixel charge coupled device (CCD) camera. Two gratings were used, giving a spectral resolution of 0.5 nm or 0.2 nm in the case of high-resolution spectra. The accelerating voltage of the electron beam for the CL spectrum excitation was set at 30 kV. During the registration, the microscope scanning mod was switched off.

The Raman spectra were recorded by a confocal micro Raman spectrometer InViaRaman in the backscattering geometry in the air at room temperature. A long working distance ×50 objective with numerical aperture NA = 0.50 was used. A series of Raman spectra using micro Raman of the same reproducibility and the signal-to-noise ratio were made. The laser spot size was smaller than the particular investigated crystal faces, and the spectra were recorded with spatial resolution of 1 μm. The argon laser green line (488 nm) was used as the excitation source. The used laser power beam was set at the power of 1 mW. For data collection and determination of the Raman spectra parameters, the Renishaw WiRE 3.4 software was used.

## 3. Results and Discussion

The HF CVD method of diamond synthesis is effective for obtaining separate microcrystals with well-developed crystal faces and diameters up to 20 μm. These microcrystals are multiple twinned. It can be caused by strongly anisotropic growth resulting in crystallites’ shape variation having equivalent faces or differences in growth rates of the crystallites bounded by different faces where the growth rate ratio depends on the growth process parameters. According to Hofmeister [13,14], multiply twinned particles (MTP) consist of several subunits. This means they are composed so that subunits of the similar shape of regular tetrahedra are twin-related, resulting in polyhedra of unique morphology and symmetry. According to Matsumoto et al. [15], in the case of a diamond, the growth of MTP up to a relatively large size seems to be kept by the durability of C-C bonds in the diamond.

### 3.1. The SEM Imaging

The SEM photos of the single microcrystals examined in this work are shown in Figure 1.

The crystallites shown in the photos were grown on the same substrate during the same synthesis cycle. The distance between them on the substrate was about 200 μm, which means they grew in the same growth conditions. As one can see, both microcrystals are more or less of the same size, but they differ in their orientation to the substrate. The synthesis of single microcrystallites on the Si substrate is possible if the substrate surface has not been previously subjected to mechanical treatment to introduce surface defects, which are the nucleation centers of the diamond phase. In that case, it is possible that diamond nucleates in the gas phase and the diamond embryos are probably caged compounds of carbon (adamantan, cuboctahedron, octane, etc.), which have weak interactions with the silicon surface. Such embryos’ structures can lead to the formation of the twins, micro twins, stacking faults, and other structural defects during the further growth of single diamond microcrystals [16]. It was already shown [17] that growth on (111) facets always leads to the inclusion of twins and micro twins, while change on (100) aspects prevents the formation of twins and stacking faults within the grains. Thus, to grow the excellent quality diamond layers, i.e., free of twins and stacking faults, the growth of (111) facets should be blocked.

### 3.2. The CL and PL Spectroscopies

Among numerous defects in diamonds, part of them can be optically active (color centers) and can be visible in CL or PL spectra. Recently, nitrogen-related defects, such as the NV center, have been of particular interest because it is the most promising for nano-photonics and quantum computing applications [18]. Nitrogen is an atom widespread in diamond, and many studies have been focused on characterizing and understanding the properties of different types of N-related defects, including the single substitutional nitrogen impurity, labeled as C—centre. The most widely studied defect in nanocrystalline diamond is the paramagnetic nitrogen–vacancy complexes (N–V) [19,20].

The CL and PL spectroscopies are the most effective methods used to study defects in both single diamond crystals and diamond polycrystalline films as well. A more precise interpretation of the obtained CL and PL spectra for investigated diamond microcrystals is presented in Figure 2. These spectra are compared to the one obtained for a single diamond monocrystal. However, the spectra are characterized by wide bands with sharp peaks. This proves the presence of color centers with different decay times. It has been found that for broadbands, it is comparable with the reciprocal value of a frequency at which this band occurs. For narrow ones, this characteristic time is at least one order shorter. The observed color centers are summarized in Table 1 together with possible luminescence peak assignments.

For analyzing the obtained results, it is convenient to divide them into two spectral areas, i.e., the 1.6–2.4 eV spectral region where color centers are visible both in the PL (2.54 eV due to excitation energy) and the CL spectra and in the 2.4–4.0 eV spectral region where color centers are recorded only in the CL spectroscopy. A strong peak in the PL spectra for microcrystallites is observed related to the GR1 defect centered at 1.675 eV. It is thought to consist of a vacancy in a neutral (V${}^{0}$) state [21]. The nitrogen vacancy related defects are among the essential ones in diamonds due to potential photonics applications. They can exist in two configurations [24]: nitrogen-neutral vacancy (NV${}^{0}$) observed at 2.156 eV in CL and PL spectra, and negatively charged nitrogen vacancy (NV${}^{-}$) at 1.945 eV. The last one is of the most important centers in diamond (possible Qubit). It may be observed in photoluminescence but does not exhibit cathodoluminescence. In the first spectral region, the peak at 2.33 eV is frequently observed in CVD diamond films, probably associated with a single-substitutional N together with a certain point defect [26]. In the case of the HF CVD microcrystals, the CL spectra taken from the (111) plane show a peak at 2.55 eV, Figure 2c. It is a naturally occurring optical defect of uncertain structure, sometimes attributed to substitutional oxygen, in type I diamonds [28]. The (111) plane in the CL spectrum shows a strong band at 2.815 eV, Figure 2e. It is very weak in the case of the (100) plane, Figure 2c and is absent in the case of the (100) oriented diamond monocrystal, Figure 2a. This CL emission is commonly known as the A-band. It is the broad peak in the undoped CVD diamond and has originated not associated with donor–acceptor recombination pairs as it is in the natural diamond [30]. However, it can be attributed to the lattice disorder such as dislocations [11] suggested earlier by J. Ruan et al. [29].

### 3.3. The Raman Spectroscopy

Raman spectroscopy is a useful method to characterize the different defect types present in diamond samples. The specific dopant introduction and defects in a diamond sample can strongly impact the lattice’s diamond quality and strain. If one characterizes only the crystalline structure of diamond, two following parameters, i.e., position $\nu_{0}$ of Raman and the FWHM of line, are of the most importance.

Our Raman measurements presented in Figure 3 reveal the dependence of the FWHM of the diamond Raman spectra taken from different crystal’s faces. If the Raman line of diamond shifts towards values above or below $\nu_{0}$ (for diamond monocrystal), this indicates that the studied sample is under compressive or tensile stresses, respectively. The residual stress, σ, can be calculated from the following relationship [31].
(1)$$\sigma =-1.08\frac{\mathrm{GPa}}{{\mathrm{cm}}^{-1}}\left(\nu_{m}-\nu_{0}\right)$$
where: $\nu_{m}$ is the measured diamond peak position for the investigated one. It is assumed in the present work that the peak position at $\nu_{0}=1330.6\;{\mathrm{cm}}^{-1}$ of the Element Six crystal indicates stress absence. The maximal shift of the Raman spectra for other microcrystallites points out the stresses occurring in them. In our case (see Figure 3), in microcrystals, regardless of their orientation, compressive stresses of approx. 3 GPa occur.

The FWHM is a measure of the crystal lattice degree disturbance. The greater the concentration of structural defects, the greater the FWHM. The defects restrict phonons’ free path or their lifetime, the length of which is proportional to the reciprocal value of the FWHM. According to the following simple relation [32]
(2)$$FWHM\times L=90\;({\mathrm{cm}}^{-1}\mathrm{nm})$$
where the FWHM is expressed in (cm${}^{-1}$) and phonon free path *L* in (nm). If one assumes that the *L* of the phonon-free path corresponds to the average distance between defects, their concentration can be estimated as proportional to $L^{-3}$.

This work aimed to investigate defects on diamond microcrystallites’ various crystallographic planes and compare their types with those observed for a single diamond crystal with (100) oriented planes. It should be noted the similarity of both CL and PL spectra for the single crystal Element Six and the CVD micrystallite recorded for the planes (100), Figure 2a–d. They point out that both the CL and PL spectra taken from the same crystallographic planes show similar color centers, mainly associated with NV type centers of different configurations. In the PL spectrum microcrystallite case, a strong peak with the energy of 1.675 eV (the GR1 defect) is also observed, associated with the vacancy, which was not recorded for the single crystal. Summing up the performed research, it can be said that the CL spectra in the 1.6–2.3 eV energy range are associated mainly with nitrogen-related defects and in the 2.3–2.9 eV energy range rather with structural ones. In Figure 3, we see the Raman peak shift which reveals the compressive stress of approximately 3 GPa. The spectrum taken from (111) planes is almost twice broader in comparison to that recorded from (100) planes, for comparable crystal sizes, which designate that the (111) crystal planes are more defective. Defect estimated concentrations for planes (111) and (100) are approximately $3.5\times 10^{18}\;{\mathrm{cm}}^{-3}$ and $8.3\times 10^{17}\;{\mathrm{cm}}^{-3}$, respectively. They differ by almost an order of magnitude. Our findings are consistent with the results of Butler et al. [33]. Based on the TEM studies, we see that (111) crystal planes are highly defective, containing a high density of stacking faults and dislocations, while (100) planes have a much lower density of such defects. The above is also confirmed by our results, i.e., the FWHM of the diamond peak in the Raman spectrum and the presence of a strong peak with the energy of 2.85 eV attributed to structural defects.

## 4. Conclusions

The diamond microcrystals of different orientations were grown by HF CVD method from the methane/hydrogen gas mixture. The SEM and combined CL, PL, and Raman spectroscopies were used to investigate the color center in (100) and (111) crystal planes. The point defects associated with N–V systems were recorded in the CL (lower energy region) and PL spectra. The A-band in the CL spectrum at 2.815 eV is the emission band in the blue-violet region. It occurs in the case of the (111) microcrystal plane. However, its intensity becomes very weak for microcrystals with (100) preferred orientation, and it is even absent in the case of the diamond monocrystal with (100) orientation. The Raman spectra analysis resulted in information about the structural effects produced in diamond microcrystals. The position shift of the diamond Raman peak indicates that microcrystals are under compressive stress. The elevated value of the FWHM means the (111) crystallographic planes contain almost an order higher defect concentration than the (100) planes. The obtained results confirm the CL blue-violet A-band can be directly related to dislocations in the diamond crystal lattice.

## Figures and Tables

**Figure 1 materials-13-05446-f001:**
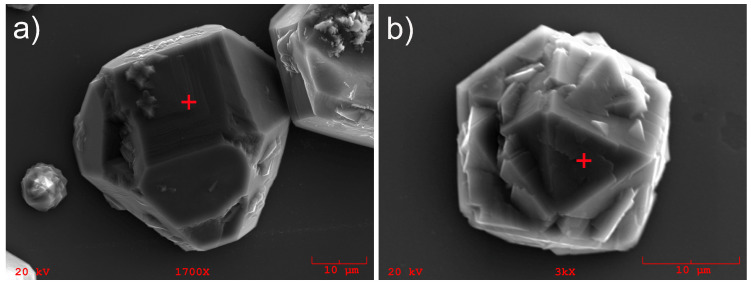
Diamond microcrystals: (**a**) (100) plane, (**b**) (111) plane. The + sign indicates places from which the cathodoluminescence CL spectrum were taken.

**Figure 2 materials-13-05446-f002:**
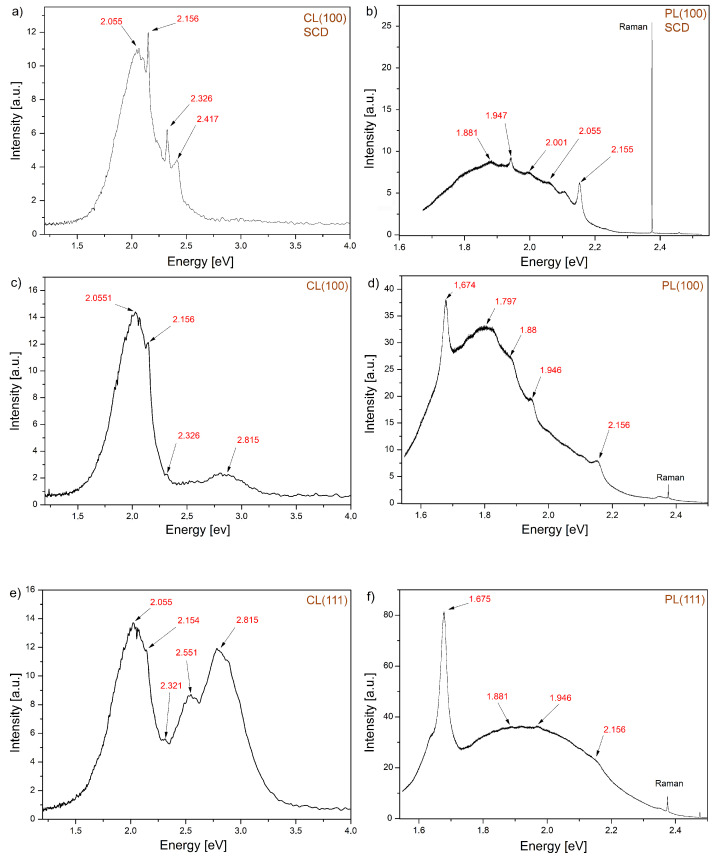
The recorded CL and photoluminescence (PL) spectra respectively for: (**a**,**b**) Element Six monocrystal, the spectra were taken from the plane (100); (**c**,**d**) hot filament chemical vapor deposition (HF CVD) microcrystal, the spectra were taken from the (100) plane; and (**e**,**f**) the spectra were taken from the (111) plane.

**Figure 3 materials-13-05446-f003:**
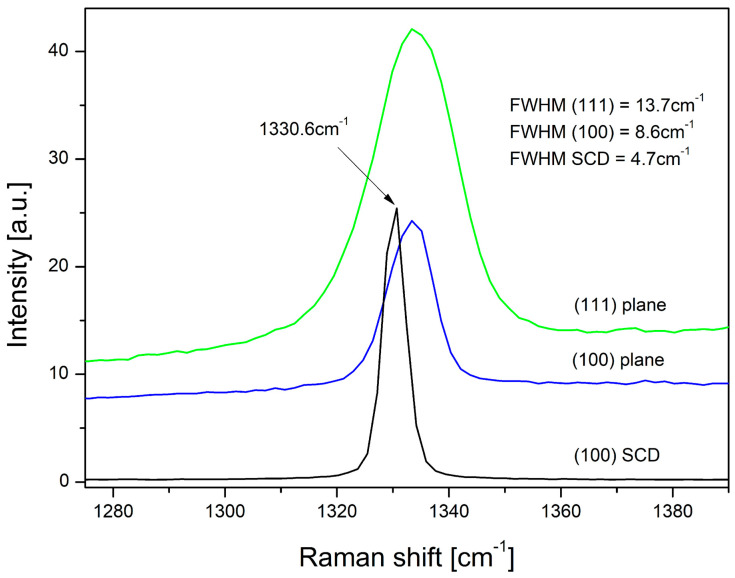
The Raman spectra for Element Six diamond monocrystal, and spectra taken from (100) and (111) planes of HF CVD diamond microcrystals.

**Table 1 materials-13-05446-t001:** The observed color centers in the CL and PL spectra.

Peak Assignment	Energy [eV]	Element Six with (100) Plane	HF CVD with (100) Plane	HF CVD with (111) Plane
GR1 is thought to consist of a vacancy in a neutral charge state (V${}^{0}$) [21]	1.675	No	Yes (PL)	Yes (PL)
Center observed in synthetic diamonds. It relates to nitrogen and is especially strong in deep yellow nitrogen- containing sectors of diamonds [22]	1.797	No	Yes (PL)	No
This defect can strongly affects the thermal conductivity of synthetic diamonds [22,23]	1.881	Yes (PL)	Yes (PL)	Yes (PL)
Nitrogen-linked centers can exist in two configurations with negative vacancy (NV${}^{-}$) at 1.945 eV and neutral one (NV${}^{0}$) at 2.156 eV [24]	1.947	Yes (PL)	Yes (PL)	Yes (PL)
H3 (NVN)${}^{0}$ and possibly N3 (3N${}^{+}$V) centers observed in natural type I diamonds [25]	2.001	Yes (PL)	No	No
This center is attributed generally to N-aggregates in A centers [8,25]	2.055	Yes (PL, CL)	Yes (CL)	Yes (CL)
Nitrogen with neutral vacancy (NV${}^{0}$) [26]	2.156	Yes (PL, CL)	Yes (PL, CL)	Yes (PL, CL)
In CVD diamond it results in pink coloration [27]	2.418	Yes (CL)	No	No
Defect of uncertain structure (some- times attributed to substitutional oxygen) in type I diamonds [28]	2.551	No	No	Yes (CL)
A-band attributed to lattice disorder such as dislocations [11,29]	2.815	No	Yes (CL)	Yes (CL)

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
