# Peer review of "Orientation Dependence of Cathodoluminescence and Photoluminescence Spectroscopy of Defects in Chemical-Vapor-Deposited Diamond Microcrystal"

_materials, 2020, doi:10.3390/ma13235446_

Round 1
Reviewer 1 Report
Dear Authors,
The issue of the submitted manuscript is interesting. You have already some relevant experiences with the topic. However, I can not recommend publishing the paper in its present form. First of all, the quality of the paper must be improved, not only the content but also the form (Why did not you use the common organization of the manuscript?). The paper is quite simple, nevertheless, its understanding is not easily traceable. I think you should explain much more about the used techniques as well as the obtained data. Much more, taking into account that the topic is not new at all.
Some of my suggestions/comments/questions:
Please, take care of how (where through the text) are you putting figures and tables.
I can only recommend to present firstly experimental part (Materials and Methods) and after that discuss the obtained results. The reorganization of the paper could help to define better the investigation line (with all interesting/important estimated values/parameters) which should be followed easily by the reader.
For example, it is not clear how you estimated the Defect concentrations.
You present the abbreviations part. Please, include there all the abbreviations used along the text.
What laser power was used for the Raman measurements? Did you observe/check the possible degradation/effect of the applied laser on the recorded Raman bands? Is it important to count on this effect? Maybe it should be interesting to discuss it.
Author Response
Dear reviewer
Thanks for your suggestions. According to your proposal, I've decided to reorganize the manuscript and the subsection Results and Discussion are followed by the Materials and Method one. In it, we've refilled other experimental details to explain the way how the investigation was prepared. The structure of the manuscript was due to the MDPI template. In my opinion, the Abbreviation subsection is a good practice for MDPI manuscript, However, all used abbreviations were introduced along the text body as well.
Both the PL and the Raman spectra were obtained upon excitation with 488 nm light from a tuneable Ar ion laser working at the power of 1 mW.
Diamond is very resistant to any radiation type. The used laser radiation of 1 mW is not able to make any structural changes.
Maybe the topic is not quite new. But we were able to find the defects concentration by exploring the simple relation. The defect concentration can be estimated using the formula
FWHM × L = 90[cm-1nm]
The FWHM of the diamond Line is expressed in [cm-1] and phonon free path L in [nm]. If one assumes that the L the phonon-free path corresponds to the average distance between defects, their concentration can be estimated as proportional to L-3.
Reviewer 2 Report
Title: Orientation dependence of cathodoluminescence and photoluminescence spectroscopy of defects in chemical-vapor-deposited diamond microcrystal.
This manuscript is about the investigation of defects in single diamond microcrystals grown by hot-filament CVD using Raman spectroscopy (RS), photoluminescence (PL), cathodoluminescence (CL), and scanning electron microscopy (SEM). Main results are;
- The CL spectra in the 1.6–2.3 eV energy range are associated mainly with nitrogen-related defects and in the 2.3–2.9 eV energy range rather with structural ones.
- Broader Raman peak from (111) planes than (100) planes mean that the (111) crystal planes are more defective.
- The CL band with an energy of 2.85 eV (A band) is mainly observed for the (111) plane, from this they claim that structural defects are the source of this type of luminescence.
How many crystallites of (100) surfaces and (111) surfaces did the authors compared in their CL and Raman measurements? Some statistics should be mentioned in the text.
Putting labels in Figure 2 like a) “CL(100) SCD”, b) “PL(100) SCD”, c) “CL(100)”, d) “PL(100)”, e) “CL(111)”, f) “PL(111)” would help the readers to understand the figure easier.
Author Response
Thanks for your comments we are trying our best to improve the manuscript. Regarding your first remark due to the statistic we’ve decided to add to the text body the following sentences in subsection Materials and Methods:
We chose diamond microcrystallites of different orientations for our research in order to study the differences in the degree of their defectiveness [8] and possible suitability for sensory applications [12]. The properties of diamond microcrystallites obtained by the HF CVD method were compared with those of a single diamond crystal purchased from Element Six company
- Paprocki, K.; Fabisiak, K.; Łoś, S.;Winnicki, J.; Malinowskii, P.; Fabisiak, R.; Franków,W. Morphological, cathodoluminescence and thermoluminescence studies of defects in diamond films grown by HF CVD technique. Optical Materials 2020, 99, 109506. doi:10.1016/j.optmat.2019.109506.
- Łoś, S.; Paprocki, K.; Fabisiak, K.; Szybowicz, M. The influence of the space charge on The Ohm’s law conservation in CVD diamond layers. Carbon 2019, 143, 413–418. doi:10.1016/j.carbon.2018.11.043.”
From our viewpoint, it is hard to say how many samples have been investigated precisely. However, we would like to ensure you that the relevant statistics have the appropriate level of certainty
Regarding your second remarks, we’ve added appropriate label to each section of Figure 2.
Reviewer 3 Report
The paper “Orientation dependence of cathodoluminescence and photoluminescence spectroscopy of defects in chemical-vapor-deposited diamond microcrystal” is a complex study of three types of diamonds. SEM, CL, PL and Raman spectroscopy are used in the research.
The paper does possess some novelty, however, I recommend major revising, it cannot be published in its present form.
My questions and recommendations are the following:
- Authors claim that the only limitation of diamonds wide application in electrical devices is a high defectivity. What about the price? It is an obvious limitation, but authors do not mention it.
- What is the exact reason for all these studies? You study defects in diamonds by all these methods to achieve what? Authors should state this clearly. It can be dependence of obtaining conditions on type and position of exact defects, or their study for the further application of diamonds with different types of defects.
- The “Method” part is located wrongly; it should be right after the introduction part. The Method should contain all used materials (diamonds), their purity and manufacturer. Maybe a more precise description of their obtaining technique or a reference to one. It seems that authors obtained diamonds on their own – then they MUST provide growing details and conditions. All used equipment should be thoroughly described. For example, “Raman spectra were measured using Raman monochromator ABC123 (Ramanour Ltd, Czech Republic). The shooting conditions were… The temperature was... A and B programs were used to process raw obtained spectra”. Otherwise reproducibility of the results are under a great doubt. Authors have to indicate details for SEM images, CL, PL, Raman spectroscopy, etc.
- The heading of the Figure 2 does not make any sense. Authors write “The recorded CL and PL spectra on left and right column respectively” Which is what? Usually it is labeled like “a, c, e – CL spectra, b, d, f – PL spectra”. Authors write “Part (a) and (b)) for Element Six monocrystal the spectra were taken from the plane (100), p” – What is Element Six monocrystal? Why use the word “part”? Authors wrote “part (c) and (d) for HF CVD microcrystal presented in Figure 1a the spectra were taken from the (100) plane” – I cannot understand this. The sentence at least lacks a comma.
- From the figure 2 I have found out that there probably are 2 types of Diamonds crystals - Element Six and HF CVD. This has not been stated anywhere above. These samples should be described in the Method.
- After the Figure 2 there is a text “This proves the presence of color centers with different decay times.” I do not see any discussion or comparison of different types of diamonds data. It is not an obvious knowledge. It also is unclear, how the decay time is connected with Fig. 2.
- In paragraph 2.3. the sentence after the Fig. 3 seem to miss the beginning.
- The sections “Discussion” is too small and poor to be a separate part of the paper.
- The paper lacks Funding sources. Even if no specific grants or scholarship were granted for this study, authors for sure did receive salary and used equipment provided by some university or research center. It can be scientific program of a Department, University or even Ministry.
- After all reference there is a sentence “Sample Availability: Samples of the compounds ...... are available from the authors”. What does it mean?
Author Response
I would like to thank you for your critical look at the manuscript. I believe your comments will help to improve its quality. Below there are answers to each of your remarks.
Regarding.1.
according to https://diamondproducers.com/app/uploads/2019/06/DPA_VALUE-truths_350dpi-no-crops_15.pdf, 1ct laboratory-grown diamond monocrystal is about 20% of the price of a natural one.
Regarding.2
to the body of the manuscript has been inserted the following text:
Optical and optoelectronic applications of CVD-grown diamonds are expected to take advantage of this material's desirable intrinsic properties. The optical properties and electrical transport properties of a wide-bandgap semiconductor such as the diamond are susceptible to states within the forbidden gap produced by lattice defects or impurity atoms. For many years, scientists have been looking for ideal candidates to use as information carriers qubits for quantum computation and communications. Nitrogen vacancy centers, NV in the diamond are among the most interesting objects for producing quantum elements that fulfill the well-established Di Vincenzo's criteria for quantum information technology at room temperature [7]. Especially an electron spins localized at N-V centers, at atomic scales can be manipulated at room temperature by applying a magnetic field, electric field, microwave radiation or light, or a combination. We believe that single crystallites and polycrystalline diamond layers with a (100) preferential orientation can be used for the above-mentioned applications. This is confirmed by the results published in our previous work [8].
7. DiVincenzo, D.P. The Physical Implementation of Quantum Computation. Fortschr. Phys. 2000,
226 9–11, 771–783. doi:10.1002/1521-3978(200009)48:9/11<771::AID-PROP771>3.0.CO;2-E.
8. Paprocki, K.; Fabisiak, K.; Łos´, S.; Winnicki, J.; Malinowski, P.; Fabisiak, R.; Franków, W. Morphological,
228 cathodoluminescence and thermoluminescence studies of defects in diamond films grown by HF CVD
229 technique. Optical Materials 2020, 99, 109506. doi:10.1016/j.optmat.2019.109506.
Regarding 3.
missing information has been inserted into the text
Regarding 4.
Proper changes have been made in Figure 2 caption.
Element Six monocrystal is a monocrystal delivered by Element Six diamond manufacturer company Element Six is s name of the company
The electron beam spot is much smaller than a dimension of a particular crystal face. With the microscope scanning mode switching off, it is possible to excite the spectrum of the CL selectively. By sign + in Figure 1 are marked approximal places in which the beam has been directed. All of these we would like to close in a short sentence "the spectra were taken from ..." Understanding that in the body text has been lack of some information, proper changes have been made in the Material and Method subsection.
Regarding 5.
The diamond single crystal was purchased from the manufacturer, i.e. from Element Six company. Diamond microcrystallites were grown in our laboratory using the HF CVD technique
Regarding 6
I agree that the knowledge of how the decay time is related to the spectra shown in Figure 2 is not apparent. However, it can be said that analyzing any spectra, the registered line width is determined by the decay time, and it intents on the transition probability. From our viewpoint, it is essential that this basic physical property of the spectroscopy method is made aware. We will probably agree with the statement that more than one article will fill this discussion. However, we've only pointed out this general relation, and the discussion of other color centers properties are presented in this one.
Regarding 7
the proper changes were introduced
Regarding 8
The manuscript has been prepared due to MDPI article template. However, according to reviews comments, it has been reorganized.
Regarding 9
the proper changes were introduced
Regarding 10
Sorry for that, the sentence has been removed.
Reviewer 4 Report
This manuscript compares the CL, PL and Raman spectroscopy measurement of HF CVD grown microcrystal on (100) and (111) facets, with the reference Element 6 (100) single crystal.
I find it difficult to gain insights and reasoning as to the presence of certain defects, as the manuscript is lack of important experimental details.
Moreover, the manuscript is not well structured and it is therefore difficult to follow the reasoning of the measurement.
(1) The impurity level (or the grade) of the Element six and the CVD microcrytstal is missing.
(2) The detail of PL is missing and therefore it is difficult to know if the authors collect the PL signal ONLY on the desired (100) and (111) facets on the polyhedron micro-crystal.
(3) Considering the color centers of diamond is very well studied, particularly the centers related to nitrogen and vacancy, the assignment carried out by the authors are rather rough and not clear.
(4) how do the authors collect Raman spectra (and PL) for a specific facet on a polyhedron microcrystal considering the collection volume? Also it can be clearly seen from the SEM image that the facets are not perfect and small islands can be seen. This would surely affect the “defective” state of Raman spectra.
Author Response
I would like to thank you for your critical look at the manuscript. I believe your comments will help to improve its quality. Below there are answers to each of your remarks.
Regarding 1
According to Element Six company the delivered monocrystal belongs to Type Ia i.e. it may contain nitrogen impurity at a level up to 0.3%. This was also confirmed by PL and CL spectrum shown in Fig.2 in the present work.
In the case of microcrystalline, they were synthesized in the HF CVD reactor by pumping down before the synthesis process to the level of 2 mbar, so there may be residual nitrogen in the reaction gases.
Regarding 2
missing information has been inserted into the text
PL measurements were made for each of the planes shown in Fig. 1 in several places, obtaining similar results. An important remark is that the accuracy of the measurements was not given, which was supplemented in the revised work.
Regarding 3
The assignment of the individual centers is given in Table 1 according to the results obtained by many researchers.
We should give the accuracy of the measurements what is done in the corrected work
Regarding 4
PL and CL measurements were made for several microcrystallites with both (111) and (100) orientations. Despite the visible secondary nucleation, the PL and CL spectra had the same character, differing only in the intensity and shift of the maxima not exceeding the value of 0.002eV. All other missing information has been inserted into the text
Reviewer 5 Report
The paper deals with the orientation dependence of defects in CVD diamond using Raman, cathode- and photo-luminescence techniques. The paper is interesting and well organized, but I have some suggestions for the authors:
- Please check and improve the English in the manuscript. I found some grammatical errors and unclear sentences: for example, the last sentence of abstract or the sentence at line 92.
For the sentence at line 92 I suggest for example: “The nitrogen’ vacancy related defects are among the essential ones in diamonds due to potential photonics applications. They can exist in two configurations [20]: nitrogen-neutral vacancy (N-V0) observes at 2.156 eV in CL and PL spectra, and negatively charged nitrogen’ vacancy (N-V−) places at 1.945 eV. The last is one of the most important centers in diamond (possible Qubit)”.
- In the introduction, the authors report that CVD diamond layers contain a high concentration of different defects but they do not mention that also the sp2 carbon component (graphite) is present in CVD diamond and plays an important role in the chemical and physical properties of this material, as reported in two papers:
(a) Velardi et al., Highly efficient and stable ultraviolet photocathode based on nanodiamond particles, Appl. Phys. Lett. 108 (2016) 083503.
(b) Velardi et al., UV photocathodes based on nanodiamond particles: Effect of carbon hybridization on the efficiency, Diam. Relat. Mater. 76 (2017) 1. - Please define all the acronyms in the manuscript: CVD at line 24, RS line 37, HF line 46 and so on.
- The author should highlight the novelty of this manuscript compared to what reported by other research groups in literature.
- It can be interesting to extend the Raman spectra of Fig. 3 up to 1800 cm-1, in order see if we have the presence of sp2 carbon content, which signal is generally found around 1580 cm-1 (G—band).
Author Response
I would like to thank you for your comments. I believe they will help to improve its quality. Below there are answers to each of your remarks.
Regarding 1
Thanks for your suggestion the unclear sentence have been rewritten.
Regarding 2 and 5
Yes, we agree that sp2 hybridized carbon can play an important role in the CVD diamond. We are aware of the G-band existence in Raman spectra. However, in this article, we want to focus on different defects types. Nevertheless, thank you for drawing my attention to the photo-optical properties of the CVD diamonds as well. We've cited both mentioned articles.
Regarding 3
The manuscript has been carefully checked and all acronyms have been introduced along with the text.
Regarding 4
A corresponding explanation is provided in the Introduction section; “In the era of miniaturization, we believe that small-size diamond microcrystallites can be used in optoelectronic devices, which is one of the motivations for the undertaking this type of research
Round 2
Reviewer 1 Report
Dear authors,
I have recognized the revised manuscript as an adequate response to the reviewer's requirements and I think that the manuscript is now acceptable for its publication in Materials.
However, I have some minor suggestions-points:
1. Pg. 3, ln.85: ...and room temperature. It was operated at room temperature. It is a repetition.
2. Pg. 3, ln. 86: The laser spot size was smaller than particular investigated crystal faces. Can you also note what microscope objective (magnification) did you use? You performed the micro measurements. How is it with the reproducibility of the Raman spectra? Did you record just one spectrum?
3. One can find a few typing errors in there. For example:
Pg. 2, ln. 63: The dot is required at the end of the sentence. Pg. 3, ln. 89: It should be Renishaw. Pg. 6, ln. 136: There is an extra space. ...
4. Please, take care of how you insert the figures and table in the manuscript. In my opinion, it is horrible if the figure or table divides the sentence, or one figure is on 2 pages although it is not necessary. It is the case of all figures and tables found in the manuscript; they could/should be inserted much better, in spaces clearly separated from the manuscript's main text.
5. It is still highly recommended to revise/check the manuscript´s English language and grammar issues.
Author Response
Thanks for your opinion about the submitted manuscript. Due to your, remarks the following changes have been done.
- The repetition has been removed.
- Lacked pieces of information regarded Raman experiment have been added.
- Punctuation imperfections have been corrected.
- I agree with you according to Figure dividing. This time it takes me a lot of time due to the Figure and the Table sizes. But now I hope I've found a proper layout.
- I've tried to do my best.
Reviewer 3 Report
Now the paper looks much better and can be published
Author Response
Thank you for your helpful remarks.